# Evolution of Therapeutic Patient Education: A Systematic Scoping Review and Scientometric Analysis

**DOI:** 10.3390/ijerph19106128

**Published:** 2022-05-18

**Authors:** Jorge César Correia, Ahmed Waqas, Isabelle Aujoulat, Melanie J. Davies, Jean-Philippe Assal, Alain Golay, Zoltan Pataky

**Affiliations:** 1Unit of Therapeutic Patient Education, Division of Endocrinology, Diabetology, Nutrition and Therapeutic Patient Education, WHO Collaborating Centre, Geneva University Hospitals, University of Geneva, 1206 Geneva, Switzerland; alain.golay@amge.ch (A.G.); zoltan.pataky@hcuge.ch (Z.P.); 2Institute of Population Health, University of Liverpool, Liverpool L69 3BX, UK; ahmed.waqas@liverpool.ac.uk; 3Centre for Health Promotion Knowledge Transfer (RESO), Institute of Health and Society, Université Catholique de Louvain, Clos Chapelle-Aux-Champs 30 B1.30.15, 1200 Brussels, Belgium; isabelle.aujoulat@uclouvain.be; 4Diabetes Research Centre, University Hospitals of Leicester NHS Trust, Leicester LE5 4PW, UK; melanie.davies@uhl-tr.nhs.uk; 5NIHR Leicester Biomedical Research Centre, Leicester General Hospital, College of Life Sciences, University of Leicester, Leicester LE5 4PW, UK; 6Fondation Recherche et Formation Pour L’enseignement du Malade, 1205 Geneve, Switzerland; jphassal@gmail.com

**Keywords:** patient education, health literacy, scientometric analysis

## Abstract

Therapeutic patient education (TPE) aims to empower the patients and their caregivers to effectively care for and manage their conditions. Such educational programs have been shown to improve health behaviors, disease outcomes, and quality of life among different patient populations. The field of TPE has evolved extensively over decades, owing to interdisciplinary research. No study so far has been done to map this field, to identify the stakeholders and gaps requiring future research. By leveraging the theory of co-citation, CiteSpace was used to visualize the bibliographic data pertaining to TPE research. A total of 54,533 articles published in English language were analyzed to identify influential funders, regions, and institutes contributing to this field. Besides these, significant theoretical and empirical contributions that shaped this field were mapped. Our analysis revealed several important insights. Most of the important theories that helped shape TPE were inspired from the social sciences. Five important research themes were identified: disorders, study designs utilized in TPE research, the scope of the TPE literature and outcomes, and populations. The research focused on improving perceptions, behaviors, and attitudes toward health promotion, reducing stigma, self-management and medication adherence. Most of the research was developed in the context of high-income countries. Future research should involve patients and use digital technology. Meta-analytical studies need to be done to identify the effectiveness and moderators of TPE interventions across different disorders. Further research should involve low and middle-income countries (LMIC) to ensure knowledge and technology transfer.

## 1. Introduction

Therapeutic patient education (TPE) helps patients, and their caregivers understand and effectively manage their chronic disorders [1,2]. These programs have been shown to alter health behaviors positively, and thus improve biological, psychological, and quality of life outcomes for many chronic disorders [3,4,5,6]. Several stakeholders and international medical organizations have recommended their use in routine patient care [4,5,7].

The field of patient education and health literacy is interdisciplinary, underpinned by frameworks derived from disciplines such as psychology and education [8,9]. It is essential to identify significant theories and empirical works on TPE, to understand how this field has evolved over the years [10]. This has become possible in recent years due to developments in the field of scientometrics (analyzing scholarly literature) and the availability of electronic bibliographic databases [11]. By utilizing statistical and machine learning approaches, analyses of bibliographic databases can yield insights about the field-specific impact of scientists, collaborative networks, institutes, and significant scholarly works [12,13]. Moreover, such exercises not only help map the scholarly contributions in a discipline but also identify gaps or areas that require more effort [14]. Thus, using these techniques, all stakeholders in the research and development process, including researchers, policymakers, and funding agencies, can determine areas where more infrastructure and human resource personnel and funding is required [13].

As previously noted, the discipline of TPE is a broad one, owing its evolution to interdisciplinary collaborative networks. Therefore, it is important to identify those influential theoretical and empirical works, which underpin this discipline. It is also essential to identify the areas where research in this field is most concentrated and the gaps where more resources are required. To our knowledge, there is a paucity of such literature in patient education. This investigation addresses this paucity, and aims to (i) quantize the research output, (ii) identify significant stakeholders, and (iii) map the influential works that have contributed to this field.

## 2. Methods

We searched the Web of Science (WoS) core database for literature on health literacy and TPE, using a pretested search strategy, from inception to May 2021 (Table 1). The present mapping study is a broad overview of the domain of TPE. In line with our research aims, we adopted a broader definition of TPE, which spans the dimensions of health promotion, prevention (selective/indicated), and treatment. It is also acknowledged that stakeholders often define TPE in a narrow context of self-management, and ongoing treatment and prevention of complications of chronic disorders [2]. We did not restrict ourselves to a particular discipline to delineate the interdisciplinary nature of the scholarly work published in this area. 

WoS yielded 54,533 articles published in the English language. Bibliographic characteristics of these articles, along with their citing references, were downloaded. This database was chosen for this investigation [11] because of its multidisciplinary coverage spanning over 20,000 journals and over 79 million reference records. It also allows advanced scientometric analyses due to the availability of cited references for each article [15].

For scientometric analyses, we utilized the CiteSpace software (v 5.7.R5W, Drexel University, Philadelphia, PA, USA), a Java-based platform for visualizing bibliographic data [16]. An excellent feature of this software is conducting co-citation analyses to reveal the collaborative networks in bibliographic data. It is based on the theory of co-citation, which posits that two studies are related when one or more studies cite them together [10,17].

We analyzed the whole dataset (without restriction to the year of publication) to quantify the trends in the research output and significant players in TPE research. However, co-citation analyses were only conducted for bibliographic records from 1990 to 2021. These records were divided into one-year periods using time-slicing methods, with each time slice represented by top 100 articles published in that period. This analytical strategy was employed for identifying top keywords, and elucidating networks of countries, institutions, research categories, and funding organizations.

Furthermore, we sought to identify influential studies in TPE research. By doing this, we envisioned constructing the roadmap presenting the evolution of the field of TPE over time. For this purpose, we divided the bibliographic records into six time periods from 1922 to 2021: pre-1985; 1985 to 1996; 1997 to 2006; 2007 to 2011; 2012 to 2016; 2017 to 2021. We chose the top 50 cited articles per one-year slice within each time-period. 

Separate network graphs were created for each time-period, which visualized different studies as nodes and links between them as lines or edges [14]. These influential studies yielded high centrality values (≥0.1) and were visualized as either a purple ring, a red ring, or a citation tree [15]. Purple rings reflect a study with a high betweenness centrality representing a new theory or concept. Red rings represent citation bursts, i.e., hot topics of research attracting high research activity in a short time. The citation tree presents the volume of citations accrued by each study, with each concentric ring corresponding to a year. The parsimony of the network was assessed using the value of modularity (Q). A high value of Q (ranging from −1 to +1) corresponds to more parsimonious networks. The influential studies identified during each period may also include studies from previous periods that may have influenced their collaborative networks [15]. 

These network analyses were run using the link reduction method with the pathfinder network scaling method. The strengths between links were estimated within slices using the Cosine index. Cluster analyses were then utilized to identify clusters of research in each time-period, where each cluster was named by processing titles of articles by employing two algorithms: log-likelihood ratio (LLR) and TF*IDF [15]. The LLR method chooses the most appropriate terms to name a cluster of research by quantifying the relationship between a term and the cluster. The TF*IDF method delineates names of clusters that are weighted by term frequencies (TF) multiplied by inverted document frequencies (IDF) [14,15,16]. The parsimony of each cluster was assessed by the number of articles in each cluster and the corresponding silhouette value. This value measures homogeneity and cohesion among nodes (studies) within a cluster and heterogeneity across different clusters [15,16,17]. 

## 3. Results and Discussion

### Publication Trends and Stakeholders on TPE 

Overall, publication trends in TPE research revealed an increasing exponential trend (Figure 1). So far, 54,533 research articles have been indexed in WoS. The earliest research study on TPE indexed in the WoS was published in the year 1922, with publications increasing exponentially after the year 1990. The research output increased from 112 in 1990 to 6004 in 2020.

Various disciplines published TPE research on public, environmental, and occupational health, yielding both the highest number of citations (*n* = 12,671) and centrality (0.22). In terms of citations, it was followed by health services research, general and internal medicine; nursing, psychology, psychiatry, education, and health policy, and oncology. In terms of innovation, however, research published in the disciplines of pharmacology and pharmacy, (bio)engineering, and economics yielded the highest centrality values (>0.10) (Appendix A).

There were 159 nodes and 1217 edges in the network graph of countries. Authors from high-income countries contributed to 97.82% of research studies published on TPE, followed by upper-middle-income countries (15.44%), low-middle-income (8.41%), and low-income countries (2.31%) (Figure 2). The USA was the single largest contributor to TPE research, accounting for 44.05% of all published research studies. It was followed by England (8.01%), Australia (7.75%), Canada (5.85%), and the People’s Republic of China (4.65%) (Table 2). Despite being upper- and lower-middle-income countries, Brazil and India were among the top-ten contributors of research in this domain. Four of the countries not only contributed to innovations in TPE research, but were also highly collaborative, as visualized by the purple rings in Appendix A and high centrality values (≥0.10).

The top 25 funding agencies (Appendix A) were in high-income countries, except for China’s National Natural Science Foundation (NSF). National Institutes of Health in USA accounted for the highest proportion of funding at 23.75% of funded research projects. Only two countries from Asia contributed to the funding of TPE research, including NSF China (0.68%), the Ministry of Education Culture Sports Science and Technology Japan MEXT (0.53%), and the Japan Society for Promotion of Science (0.45%). The European Commission ranked fourth overall (2.12%).

## 4. Keywords

Top keywords were analyzed to yield two important insights; (a) central and most cited keywords to identify essential themes of research and (b) burst words to identify hotspots of research in different time-periods. First, a critical appraisal of central and keywords accruing > 300 citations was conducted to analyze themes of research that have most frequently been researched in TPE. There were 413 nodes and 2455 edges in this network graph. It revealed five important research themes including different chronic disorders, study designs, scope, outcomes, and populations (Appendix A). Table 3 presents important chronic disorders exhibiting substantial research activity (citation bursts) during specific time-periods.

TPE research was focused on improving perceptions, behaviors, and attitudes for health promotion, self-management, and medication adherence, and reducing stigma. In addition, knowledge- and awareness-related outcomes, quality of life, and decision making for patients were frequently cited outcomes. From 2020 to 2021 specifically, keywords showed social media, health communication, and coronavirus as the top-cited keywords, indicating a shift in research to these important topics (Appendix A). Significant study design related keyword exhibiting a burst of research activity was controlled trial (1991 to 2009), while the top-five chronic conditions exhibiting the most robust research activity were asthma (the year 1991 to 2013), diabetes mellitus (1991 to 2004), hypertension (1991 to 2008), rheumatoid arthritis (1992 to 2003), and smoking cessation (1991 to 2006). 

### 4.1. Clusters and Influential Publications in TPE Research

To examine clusters of research in TPE, the research literature was divided into two timeframes: pre-1985 and post-1985. Due to the extent of literature, the bibliographic dataset of literature published after 1986 was divided into seven, five-yearly time slices. This strategy allowed us to get a dynamic snapshot of changing research themes over time. This also allowed us to identify important pieces of work published during each time slice, thus, identifying the evolution and complexity of TPE research over time (Table 4 and Appendix A).

### 4.2. Analysis of the Literature Published before 1985

Before the year 1985, 1942 research studies accounting for 3.57% of the total research output in TPE were published. The earliest study indexed in WoS was published in the year 1922. In the resulting network, there were 320 nodes and 881 edges, with a density of 0.0173.

Five research clusters with a silhouette value > 0.9 were identified (Figure 3). The largest research cluster comprised of 47 studies focused on *program planning* (ID # 0, LLR), and included studies on program planning, cost-benefit, and impact evaluation of health education programs. The first cluster comprised 38 studies and was termed *behavioral pediatrics*, which included research on school health education programs. The second cluster *consumer health education* included studies focused on TPE in chronic disorders, aging and hypertension in the elderly. Cluster # 3 *HMO environment* predominantly comprised of health education and health promotion among individuals living in houses with multiple occupancies. The last cluster ID # 5 comprised 15 studies, termed the *organized patient education program*.

Only one study during this period yielded a high centrality score of 0.21. In this study, Farquhar et al. tested the effectiveness of a novel community education intervention to improve cardiovascular health [24]. This novel intervention was a combination of an extensive mass-media campaign plus face-to-face instruction comprised of information and behavioral skills to affect attitude and motivation [25]. It also attracted a significant research burst for 4.28 years. During this period, the works of Green LW attracted significant research activity, albeit without yielding a high centrality score [26]. Notable works focused on frameworks for studying the impact and cost-effectiveness evaluations of TPE programs.

### 4.3. Analysis of the Literature Published from 1985 to 1996

From 1985 to 1996, a total of 3026 studies were published. There were 608 nodes and 1362 edges in the network, yielding an adequate modularity (0.87) and mean silhouette value (0.96). Several innovative and central works were published during this period, indicating rapid scholarly progress in TPE research (Appendix A). These influential research studies were diverse and employed rigorous study designs. All the influential works were published from high-income western nations.

There were sixteen parsimonious clusters of research (Figure 4), represented by at least ten studies published during this period. The cluster (#0) comprised of 69 studies with a mean silhouette value of 0.99. It was termed as *social skills training* (LLR) and schizophrenia (LSI). It was followed by clusters entitled *development*, *reliability*, *and prospective pricing* (#1); relevant theories (cluster #2), national asthma education (#3), therapeutic potential and rheumatoid arthritis (#4), diabetes, brief-office based intervention and dietary management (#5), asthma and school setting (#6); polio vaccine information and mammography usage (#7), AIDS prevention program (#8), health belief model (#9), and five city project and Stanford (#10). The following two clusters were entitled adolescents (#11) and learning disabilities (#12).

Several important works in psychology that underpinned the philosophical and conceptual foundations of TPE were published during this period. Lazarus and Folkman (1984) presented an integrative theoretical analysis entitled *Stress*, *Appraisal and Coping*, as a detailed theory of psychological stress which built on the concepts of cognitive appraisal and coping. These have since become major themes of theory and investigation [18]. Bandura A (1986) published their seminal work *Social Foundations of Thought and Action: A Social Cognitive Theory*, an influential piece of work in psychology, which presented a comprehensive theory of human motivation and action [19]. It analyzed the role of cognition, self-regulation, and self-reflective processes in improving psychosocial functioning. Another critical insight during this period was the concept of self-efficacy and its application in bringing about a sustained behavior change. This was elaborated by Strecher et al. (1986), who examined its utility in several domains, including cigarette smoking, obesity, contraception, alcohol use, and physical activity. Extending upon the cognitive behavioral model [27], Marlatt GA (1985) presented their breakthrough *CB model of the relapse process*. This work was seminal because it challenged the notions of viewing relapse as a “treatment failure” [20]. The field of TPE was also developed further based on Ajzen’s theory of planned behavior, which emphasized that behavioral intentions and actions result from the interplay of three components: attitudes, subjective norms, and perceived behavioral control [8]. Using path analysis techniques, Rippetoe and Rogers (1987) described the utility of protection-motivation theory on adaptive and maladaptive coping skills to manage health threats [28].

A significant body of work focused on developing psychological constructs and testing measures for utilization in TPE research. The Diagnostic and Statistical Manual (3rd edition) proved to be a major work in guiding psychoeducation programs for people with chronic mental illnesses [29]. Woodcock (1990) and McGrew (1991) laid down the theoretical foundations of -R Measures of Cognitive Ability and Anastasi A (1986) published an easy to understand book on psychological testing [21,22,30].

A body of work focused on understanding processes to improve patient participation in clinical care. However, most of these research studies focused on either cardiovascular diseases or post-surgical recovery. For instance, Davis et al. (1990) emphasized the importance of designing patient reading material with better readability [31], and Greenfield et al. (1988) tested the utility of algorithmic prompts to encourage patients to negotiate medical decisions [32]. Morisky et al. (1983) yielded the highest centrality (0.28) during this period, demonstrating the high effectiveness of the TPE program in improving pressure control and mortality among hypertensive patients belonging to poor urban households [33]. Devine et al. (1986) presented a meta-analysis of 102 studies to demonstrate the clinical effectiveness and cost-saving benefits of psychoeducational interventions among surgical patients to improve recovery, pain, satisfaction, and psychological well-being [34]. In 1983, Devine and Cook used meta-analytical methods to demonstrate lower post-surgical hospital stays among surgical patients [35]. Mullen et al. (1985) presented another meta-analysis comparing the effectiveness of educational programs for people with long-term health problems, focusing on moderators of treatment [36]. Faraquhar (1990) led the Stanford five-city project and demonstrated the effectiveness of low-cost, community-wide TPE programs in improving outcomes in stroke and coronary heart disease [25].

Only one study on a psychiatric disease yielded significance during this period. In 1986, Hogarty et al., published a landmark study combining a patient-centered behavioral treatment and a psychoeducational family intervention to reduce relapse rates in patients with schizophrenia [37].

### 4.4. Analysis of the Literature Published from 1997 to 2006

From 1997 to 2006, a total of 7370 records were analyzed. The resulting network graph comprised of 435 nodes and 713 edges: with a modularity of 0.79 and silhouette value of 0.94 (Appendix A). Eleven research clusters comprising at least ten studies and adequate silhouette values were identified. The largest health cluster (#0) focused on functional health literacy, comprising of 63 studies. Cluster #1 comprising of 50 studies focused on digital interventions. The rest of the clusters were entitled major depression (#2), health education and health promotion (#3), rheumatoid arthritis and educational-behavioral joint protection program (#4), bipolar disorder (#5), diabetes TPE research (#6 and #9), health education material and physical activity (#7), family intervention and schizophrenic patient (#8), and the American Heart Association Disease Management (#10) (Figure 5).

Most of the influential articles on TPE reported findings from controlled trials and impact assessments of TPE interventions in Western countries. However, Campbell and colleagues (2000), building on a more extensive report of the Medical Research Council, published a debate to attract the research community to go beyond RCT and focus on the design, evaluation, and implementation of complex TPE interventions [38]. In addition, three works of note focused on measuring patients’ literacy skills and designing health communication strategies for patients with low literacy levels [39,40,41,42].

Two critical pieces of legislation and policy documents provided an impetus to TPE research during this period and beyond. The guide to clinical and preventive services published by the US Preventive Services Task Force (1996), endorsed counseling interventions for prevention and clinical services including counselling for substance misuse, diet, and exercise, injury prevention, sexual behavior, and dental health [43]. While in the UK, an expenditure and funding report published by Health and Personal Social Services Programmes placed much emphasis on TPE and counseling services [44].

In mental health, the most influential work comprised two reviews of research evidence on social skills training, family interventions, cognitive rehabilitation, and coping with residual positive symptoms among patients with schizophrenia [45,46]. In addition, four studies explored multidisciplinary interventions to prevent the readmission of elderly patients with congestive heart failure [47]; quality of life in asthma and chronic obstructive pulmonary disease [48]; and type 2 diabetes mellitus [49]. This period also saw two central works utilizing a digital platform to reduce hospital admission through computer supported education for patients with asthma and to improve dietary behavior in primary care [50,51].

### 4.5. Analysis of the Literature Published from 2007 to 2011

From 2007 to 2011, a total of 7517 records were published during this period. As a result, there were 152 nodes and 226 edges in the network graph, with a mean modularity of 0.61 and silhouette value of 0.81 (Appendix A). 

Two studies focused on theoretical developments in the field of TPE. The narrative during this period shifted to health literacy, emphasizing the need to understand this complex construct. In this context, David Baker’s (2006) perspective on the meaning and the measure of health literacy yielded much significance in this field [23]. An influential review during this period by DeWalt and colleagues (2004) showed that the patients with low literacy scored poorer on several psychosocial and biological health outcomes [52].

These theoretical works aided in several empirical investigations. For example, Davis et al. (2006) demonstrated a poorer understanding of dosage labels to be a major factor for non-adherence [53]. Lower use of preventive services was also evident in people with low literacy levels [54]. Other hazards of low literacy levels were also demonstrated in people with diabetes and cancer patients [55], where a poorer understanding of cancer screening and cancer symptoms adversely affects the stage of cancer at diagnosis. The most central intervention-focused reviews and trials during this period researched bipolar disorder [56], type 2 diabetes [57], and chronic conditions [58].

Computer-assisted interventions and health informatics began to emerge as a central field of study during this period. Three significant studies were published during this period. Berland et al. (2001) demonstrated that most of the information provided on the internet might be accurate but lacks good coverage and readability [59]. Deborah et al. (2003) reported computer-delivered TPE to be effective [60]. There were barriers to its access, but generally, no socioeconomic disparities in its usage were evident. Ziebland et al. (2004) conducted a qualitative study among patients with cancer and explored their internet usage in context of their illnesses [61].

There were five clusters with ≥10 studies and adequate silhouette values. Cluster #0 comprised 22 studies and was labelled as clinical association and health literacy. Cluster #1 was labelled as screening question and health literacy, cluster #2 as risk factor, self-management, and controlled trial, cluster #3 as qualitative evaluation and controlled trial, cluster #4 as computer literacy, and cluster #5 as bipolar disorder and psychosocial treatment (Figure 6).

### 4.6. Analysis of the Literature Published from 2012 to 2016

From 2012 to 2016, a total of 13,746 studies were published. There were 150 nodes and 195 edges in the network graph, yielding a modularity of 0.63 and silhouette value of 0.85 (Appendix A). Research during this period diversified in terms of populations (Figure 7). In contrast to previous periods, an emphasis on racial disparities, non-Caucasians, and ethnic minorities was evident. The largest cluster (#0) comprising 20 studies pertained to health literacy decline and health disparities. The second cluster (#1) focused on readability assessment of internet-based education material. The third cluster (#2) was on oral health and mental health literacy. It was followed by health literacy questionnaire (#3), health literacy (#4), medication adherence (#5), ethnic group (#6), cohort study (#7), and health outcome (#8) (Figure 7).

By examining the most central works on TPE during this period, three major themes of research studies emerged. The first body of central works focused on the evolving concept of health literacy, its definitions, and its measurement methods. Don Nutbeam (2008) emphasized the evolving concept of health literacy by dividing it into two disciplines: (i) a measure of clinical risk emphasized in public health and (ii) as an asset emphasized in education research into adult learning and health promotion [62]. The *Newest Vital Sign* emerged as a widely used short tool for the measurement of health literacy [63,64]. Further conceptual work on health literacy was conducted by Sørensen et al. (2012), who put forward an influential integrative model for health literacy to aid in the development of health-promotion interventions [65]. Finally, Jordan et al. (2010) summarized the evidence for health literacy scales and reported inconsistencies in concepts and measurements underpinning these scales [66].

This period also recognized patient activation as a measure for health-related outcomes and as a component of TPE interventions [9]. TPE intervention for diverse populations gained momentum including child-related outcomes [67], and geriatric populations [68,69]. The seminal work by Osborn et al. (2011) emphasized that health literacy explains racial disparities in health outcomes [70].

### 4.7. Analysis of the Literature Published from 2017 to 2021

From 2017 to 2021, a total of 21,185 articles were published. The resulting network had 194 nodes and 288 edges. The modularity of the network was 0.66 and silhouette value 0.89 (Appendix A). Nine parsimonious clusters with >10 studies and adequate silhouette values were identified during this period. Cluster (#0) was entitled pragmatic randomized controlled trial, followed by e-health literacy scale (#1), COVID-19 pandemic (#2), health literacy questionnaire (#3), study design and latent trait (#4), mixed methods study (#5), health literacy and decision aid (#6), mental health literacy (#7), and diabetes mellitus (#8) (Figure 8).

A total of 19 studies yielded a centrality value > 0.10 during this period. Three policy documents in this period attracted research on TPE. The diabetes atlas (IDF, 2017) published by the International Diabetes Federation and the WHO’s Global Report on Diabetes (2016) yielded much significance among TPE researchers, and introduced several initiatives and calls for TPE programs targeting people with diabetes mellitus [71,72]. Among the research community, a focus was observed on measuring health literacy in different regions by conducting extensive surveys. Sorensen et al. (2015) published the European Health Literacy Survey to report literacy levels in eight European countries [73]. They reported a social gradient in health literacy levels with poorer literacy observed among people with financial deprivation, low social status, low education, and old age. These findings were also corroborated by Rikard et al. (2016), who reported health literacy disparities in the US [74], Levin-Zamir et al. (2016) in Israel [75], and Duong et al. in Taiwan [76]. During this period, Duong et al. (2015) also validated the European health literacy questionnaire (HLS-EU-Q47) in six Asian countries [77].

This period also saw the publication of several influential articles on health literacy assessment tools. An influential review by Altin et al., reported that there was a lack of clear consensus on health literacy measurements [78]. In a similar context, a review of 51 instruments for measuring health literacy [79] highlighted that these tools represent only a narrow set of conceptual dimensions, limited modes of administration, and lack information on their reliability and validity. Meta-analytical studies published during this period investigated the negative association between health literacy and the ability to evaluate online health information (Divani et al., 2015); self-management skills in chronic disease management [80]; medication adherence [81,82]; and the role of decision aids in choosing health treatments or making screening decisions [83].

Only one study on mental health gained adequate centrality during this period. However, in contrast to previous interventions, it was focused on mental health first aid to empower the public to approach, support, and refer individuals in distress [84].

The year 2020 to 2021 was unique because the research focus shifted to the COVID-19 pandemic. There was a mushroom growth in publications during this period. After the WHO declared the COVID-19 as a pandemic, a lot of researchers strived to work for solutions. Three articles of interest were published during this period. Nguyen et al. (2020) reported that people with suspected COVID-19 symptoms reported a higher likelihood of depression and quality of life. However, this effect was moderated by health literacy levels [85]. Wang et al. (2020) reported the psychological impact of COVID-19 in 194 cities in China. A high proportion of Chinese people reported moderate to severe psychological impact from the COVID-19 pandemic and one-third reported moderate to severe symptoms of anxiety [86]. Finally, Brooks et al. (2020) published a rapid review on the psychological impact of quarantine and recommended that officials should quarantine individuals for no longer than required, provide clear rationale for quarantine and information about protocols, and ensure sufficient supplies are provided [87].

## 5. Discussion

The present scientometric analysis is part of a larger project the research team is leading to evaluate the state of research in patient education and its effectiveness in managing chronic disorders. We found that much significant research works in TPE and outcomes tested, lacked the involvement of patients. Patient and lay public involvement in health research is a vast field at present. Just as they are important stakeholders in clinical decision-making, their involvement in the design of TPE interventions should be ensured. This can lead to more feasible, acceptable, sustainable, and transparent interventions [88]. 

More recently, much focus has been put on the use of digital technology in the provision of healthcare [89]. The adoption of digital technology is a massive breakthrough in population health because it can ensure equitable health access to specialized resources [90,91]. Although digital health has been tested in several facets of healthcare, including access to specialized clinical help [89], more research is needed in providing TPE. Currently a lot of research is going on in the provision of TPE using web platforms, which needs to be streamlined by government stakeholders to prevent content that may cause harm. The recent COVID-19 infodemic of hoaxes and conspiracy theories is an excellent example in this context [92].

Although the publication trends in TPE seem satisfactory, a considerable research deficit was seen in low- to middle-income countries (LMIC). This may be partly explained by a lack of clinical resources, including healthcare personnel (HCP) and medications [93,94]. The shortage of HCPs is far below the recommended numbers by the WHO, leading to short consultation times, and therefore, patients with chronic disorders usually get poorer (if any) TPE [95,96]. The world, therefore, is far from achieving equitable health access as declared in the Alma Ata declaration four decades ago [97]. To counter this inequity in resources, multidisciplinary research is required in these countries to develop culture-sensitive theories and intervention programs and innovations for equitable delivery. Researchers based in high-income countries should collaborate with those in LMICs to ensure knowledge and technology transfer. The knowledge transfer activities could range from training and education of scholars, assistance in the development of interventions, and cross-cultural adaptations of important theories for conceptual advances. 

The advances in research in TPE should be centralized on a single platform, gathering researchers and stakeholders from around the globe. Priority setting should be done for the gaps identified to yield maximum benefits. This centralized platform could be housed as a collaboration between different international societies on TPE. An important example to emulate could be the James Lind Alliance for priority setting in clinical trials, which involves patients and physicians setting priorities for research in different disorders. By using Delphi study methods, a stakeholder consensus could be achieved in TPE research. Another critical task for this consensus committee should be the standardization of definitions and terminologies across the TPE field. This would allow the harmonization of literature and concepts across different study disciplines.

Meta-analytical studies need to be done to identify the effectiveness of TPE interventions across different disorders and define their moderators. For instance, it is important to identify in which conditions, modalities of delivery and for whom these TPE interventions work the best. This is indeed possible by using meta-analytical methods. In addition, research is required on large-scale implementation aspects of TPE interventions. Many clinical trials and meta-analyses have been conducted to prove the effectiveness of these interventions across a variety of biological and psychosocial outcomes. However, the real-life implementation of any intervention and its feasibility and uptake are determined by various forces, which could be identified with implementation-driven studies.

## 6. Conclusions

The present scientometric analyses reveal several essential insights into TPE research. First, research in TPE is very diverse, with research topics ranging from the development of theories and interventions, development and testing of tools to measure outcomes and tailoring and testing of TPE interventions. Second, TPE is a highly interdisciplinary area, with various interventions underpinned by theories from intelligence research, education, and psychology and public health. Third, restricted research interests on specific chronic disorders were evident in this scientometric analysis. A satisfactory trend in research activity was observed over the years; however, most of this activity was focused on high-income countries. Low- to middle-income countries only account for a small proportion of this research activity.

There are several strengths of this study. Firstly, using reproducible and robust scientometric techniques, it provides a thorough overview of the influential stakeholders in TPE research. In addition, a summary of landmark scholarly works which have contributed to the progress of this field has been provided for the readers. Another strength of this study is the use of an elaborate search strategy using bibliographic data from the Web of Science database, which allowed us to visualize the interdisciplinary nature of the field of TPE. This is important to identify the theories and empirical works cited in TPE that have been borrowed from other disciplines and thus, act as “bridging nodes” between different disciplines. 

An important limitation of this study is the restriction to peer-reviewed publications published in the English language. Analysis was restricted to the English language to avoid computational errors in the scientometric analyses. However, this may have resulted in some degree of information bias where landmark studies published in languages other than English may have been omitted. Besides, restricting our analyses to peer-reviewed publications may have led to omissions of landmark non-peer-reviewed working papers commissioned by stakeholders such as the World Health Organization. Therefore, we encourage researchers to consider these limitations in designing future investigations.

## Figures and Tables

**Figure 1 ijerph-19-06128-f001:**
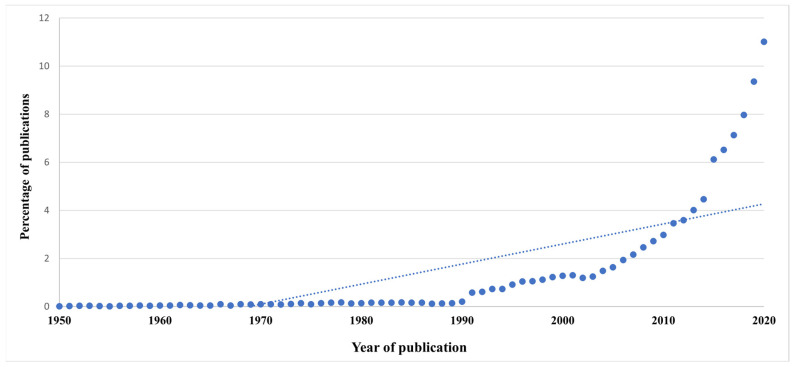
Trends of publications in patient education research.

**Figure 2 ijerph-19-06128-f002:**
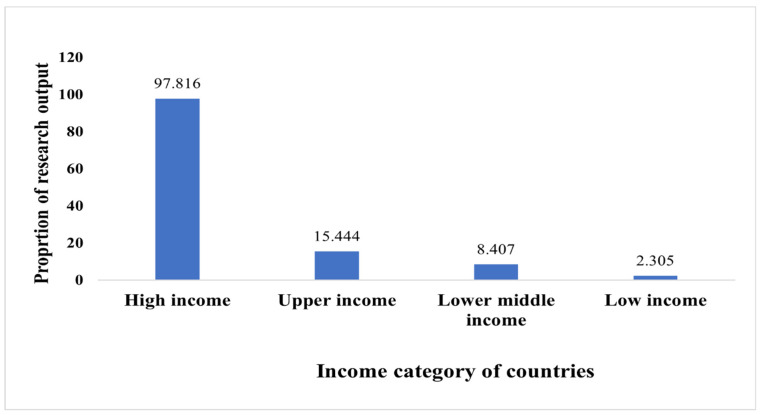
Regional contributors of patient education research (includes multi-authored studies).

**Figure 3 ijerph-19-06128-f003:**
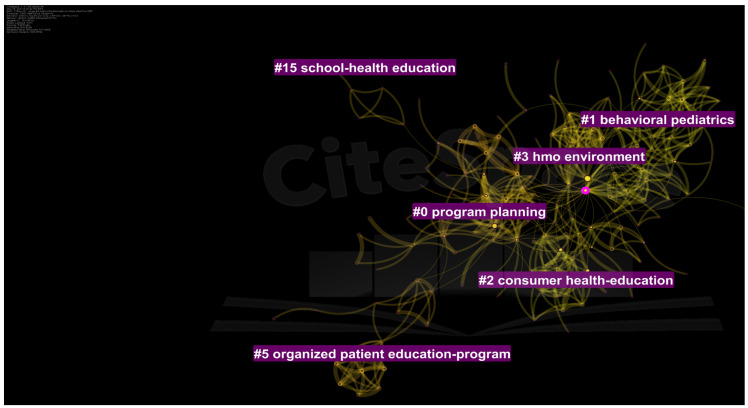
Top clusters of research in patient education research before 1985.

**Figure 4 ijerph-19-06128-f004:**
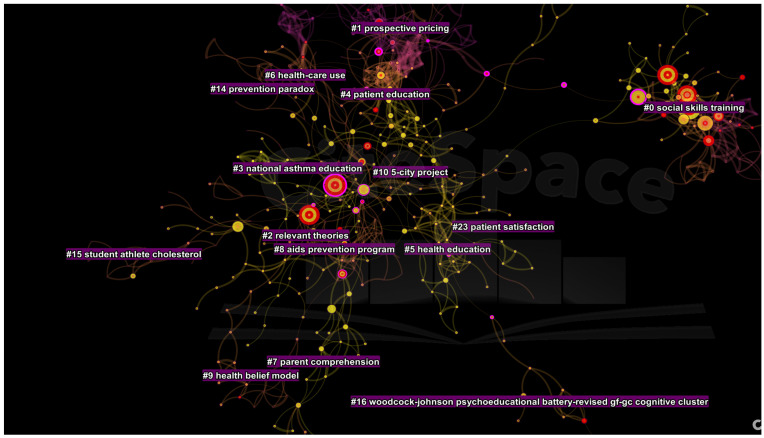
Top clusters of research in patient education from 1985 to 1996.

**Figure 5 ijerph-19-06128-f005:**
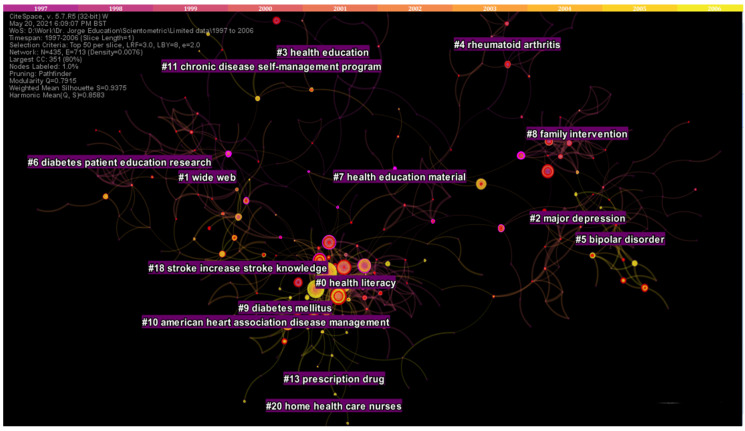
Top clusters of research from 1997 to 2006.

**Figure 6 ijerph-19-06128-f006:**
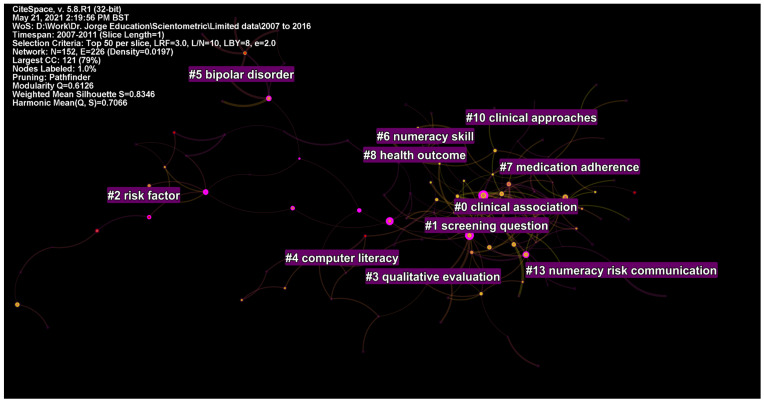
Top clusters of research from 2007 to 2011.

**Figure 7 ijerph-19-06128-f007:**
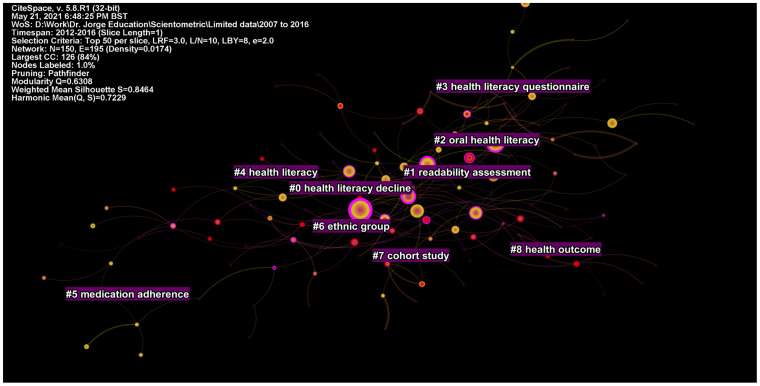
Top clusters of research published from 2012 to 2016.

**Figure 8 ijerph-19-06128-f008:**
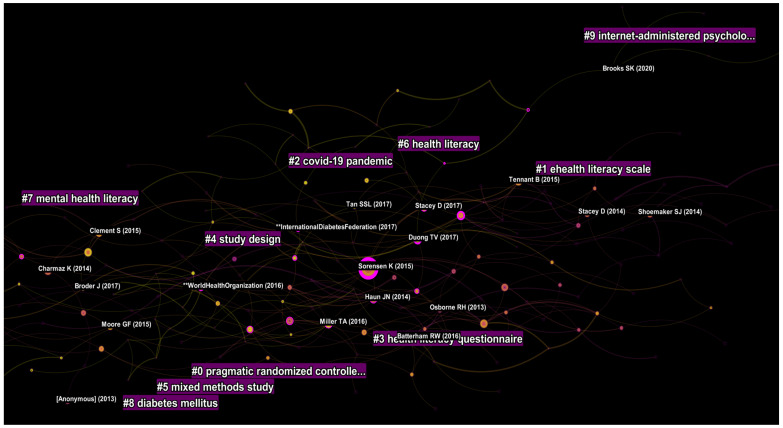
Top clusters published from 2017 to 2021.

**Table 1 ijerph-19-06128-t001:** Search strategy to identify studies pertaining to patient education research.

Scope	Details
Terms	(“Health education*” OR “patient education*” OR psychoeducation* OR “therapeutic education*” OR “consumer health information*” OR “health knowledge” OR “health literacy”)
Scope of search	TS: Topic, Title, Abstract, Author Keywords and Keywords Plus^®^
Period	Through 17 May 2021
Filters	Articles published in English language
Total search results	54,533

**Table 2 ijerph-19-06128-t002:** Top-five stakeholders in patient education research.

Institutes	Countries	Journals
University of California System (*n* = 2215)	USA (*n* = 24,021)	Patient Education and Counseling (*n* = 1109)
Harvard University (*n* = 1363)	England (*n* = 4367)	BMC Public Health (*n* = 666)
University of London (*n* = 1256)	Australia (*n* = 4226)	PloS One (*n* = 619)
University of North Carolina (*n* = 1086)	Canada (*n* = 3188)	Journal of School Health (*n* = 586)
University of Texas System (*n* = 1084)	Peoples Republic of China (*n* = 2537)	International Journal of Environmental Research & Public Health (*n* = 525)

**Table 3 ijerph-19-06128-t003:** Keyword bursts for disorders explored in patient education research.

Keywords	Strength	Begin	End	1990–2021
Alcohol	34.6	1991	2006	▂ ▃▃▃▃▃▃▃▃▃▃▃▃▃▃▃▃ ▂▂▂▂▂▂▂▂▂▂▂▂▂▂▂
Arthritis	30.51	1991	2002	▂ ▃▃▃▃▃▃▃▃▃▃▃▃ ▂▂▂▂▂▂▂▂▂▂▂▂▂▂▂▂▂▂▂
Asthma	103.39	1991	2013	▂ ▃▃▃▃▃▃▃▃▃▃▃▃▃▃▃▃▃▃▃▃▃▃▃ ▂▂▂▂▂▂▂▂
Blood pressure	53.05	1991	2008	▂ ▃▃▃▃▃▃▃▃▃▃▃▃▃▃▃▃▃▃ ▂▂▂▂▂▂▂▂▂▂▂▂▂
Cardiovascular disease	20.76	2005	2008	▂▂▂▂▂▂▂▂▂▂▂▂▂▂▂ ▃▃▃▃ ▂▂▂▂▂▂▂▂▂▂▂▂▂
Childhood asthma	5.82	1992	1998	▂▂ ▃▃▃▃▃▃▃ ▂▂▂▂▂▂▂▂▂▂▂▂▂▂▂▂▂▂▂▂▂▂▂
Cholesterol	18.22	1991	1997	▂ ▃▃▃▃▃▃▃ ▂▂▂▂▂▂▂▂▂▂▂▂▂▂▂▂▂▂▂▂▂▂▂▂
Cigarette smoking	23.95	1992	1998	▂▂ ▃▃▃▃▃▃▃ ▂▂▂▂▂▂▂▂▂▂▂▂▂▂▂▂▂▂▂▂▂▂▂
Coronary heart disease	20.72	2005	2007	▂▂▂▂▂▂▂▂▂▂▂▂▂▂▂ ▃▃▃ ▂▂▂▂▂▂▂▂▂▂▂▂▂▂
Dental caries	8.37	1990	2002	▃▃▃▃▃▃▃▃▃▃▃▃▃ ▂▂▂▂▂▂▂▂▂▂▂▂▂▂▂▂▂▂▂
Diabetes	31.4	2015	2017	▂▂▂▂▂▂▂▂▂▂▂▂▂▂▂▂▂▂▂▂▂▂▂▂▂ ▃▃▃ ▂▂▂▂
Diabetes mellitus	28.53	1995	2007	▂▂▂▂▂ ▃▃▃▃▃▃▃▃▃▃▃▃▃ ▂▂▂▂▂▂▂▂▂▂▂▂▂▂
Diarrhea	6.59	1991	1996	▂ ▃▃▃▃▃▃ ▂▂▂▂▂▂▂▂▂▂▂▂▂▂▂▂▂▂▂▂▂▂▂▂▂
Heart disease	9.17	1992	1997	▂▂ ▃▃▃▃▃▃ ▂▂▂▂▂▂▂▂▂▂▂▂▂▂▂▂▂▂▂▂▂▂▂▂
HIV infection	20.02	1992	2000	▂▂ ▃▃▃▃▃▃▃▃▃ ▂▂▂▂▂▂▂▂▂▂▂▂▂▂▂▂▂▂▂▂▂
Hospitalization	4.76	1991	1993	▂ ▃▃▃ ▂▂▂▂▂▂▂▂▂▂▂▂▂▂▂▂▂▂▂▂▂▂▂▂▂▂▂▂
Human immunodeficiency virus	12.01	1991	1995	▂ ▃▃▃▃▃ ▂▂▂▂▂▂▂▂▂▂▂▂▂▂▂▂▂▂▂▂▂▂▂▂▂▂
Hypertension	12.17	2011	2013	▂▂▂▂▂▂▂▂▂▂▂▂▂▂▂▂▂▂▂▂▂ ▃▃▃ ▂▂▂▂▂▂▂▂
Immunization	4.65	1992	1996	▂▂ ▃▃▃▃▃ ▂▂▂▂▂▂▂▂▂▂▂▂▂▂▂▂▂▂▂▂▂▂▂▂▂
Infection	7.86	2010	2011	▂▂▂▂▂▂▂▂▂▂▂▂▂▂▂▂▂▂▂▂ ▃▃ ▂▂▂▂▂▂▂▂▂▂
Low birth weight	4.07	1992	1994	▂▂ ▃▃▃ ▂▂▂▂▂▂▂▂▂▂▂▂▂▂▂▂▂▂▂▂▂▂▂▂▂▂▂
Malignant melanoma	7.16	1995	1999	▂▂▂▂▂ ▃▃▃▃▃ ▂▂▂▂▂▂▂▂▂▂▂▂▂▂▂▂▂▂▂▂▂▂
Mammography	12.54	1991	2004	▂ ▃▃▃▃▃▃▃▃▃▃▃▃▃▃ ▂▂▂▂▂▂▂▂▂▂▂▂▂▂▂▂▂
Mental health	44.63	2019	2021	▂▂▂▂▂▂▂▂▂▂▂▂▂▂▂▂▂▂▂▂▂▂▂▂▂▂▂▂▂ ▃▃▃
Mortality	4.72	1994	1995	▂▂▂▂ ▃▃ ▂▂▂▂▂▂▂▂▂▂▂▂▂▂▂▂▂▂▂▂▂▂▂▂▂▂
Myocardial infarction	12.93	1992	1998	▂▂ ▃▃▃▃▃▃▃ ▂▂▂▂▂▂▂▂▂▂▂▂▂▂▂▂▂▂▂▂▂▂▂
Oral health	39.24	2014	2018	▂▂▂▂▂▂▂▂▂▂▂▂▂▂▂▂▂▂▂▂▂▂▂▂ ▃▃▃▃▃ ▂▂▂
Pain	18.58	2013	2014	▂▂▂▂▂▂▂▂▂▂▂▂▂▂▂▂▂▂▂▂▂▂▂ ▃▃ ▂▂▂▂▂▂▂
Relapse	66.57	1991	2006	▂ ▃▃▃▃▃▃▃▃▃▃▃▃▃▃▃▃ ▂▂▂▂▂▂▂▂▂▂▂▂▂▂▂
Rheumatoid arthritis	48.38	1992	2003	▂▂ ▃▃▃▃▃▃▃▃▃▃▃▃ ▂▂▂▂▂▂▂▂▂▂▂▂▂▂▂▂▂▂
Schizophrenia	4.21	1996	1998	▂▂▂▂▂▂ ▃▃▃ ▂▂▂▂▂▂▂▂▂▂▂▂▂▂▂▂▂▂▂▂▂▂▂
Schizophrenic patient	16.54	1992	1999	▂▂ ▃▃▃▃▃▃▃▃ ▂▂▂▂▂▂▂▂▂▂▂▂▂▂▂▂▂▂▂▂▂▂
Sexual behavior	4.55	1991	1997	▂ ▃▃▃▃▃▃▃ ▂▂▂▂▂▂▂▂▂▂▂▂▂▂▂▂▂▂▂▂▂▂▂▂
Sexually transmitted disease	8.53	1991	2001	▂ ▃▃▃▃▃▃▃▃▃▃▃ ▂▂▂▂▂▂▂▂▂▂▂▂▂▂▂▂▂▂▂▂
Smoking	15.08	2009	2010	▂▂▂▂▂▂▂▂▂▂▂▂▂▂▂▂▂▂▂ ▃▃ ▂▂▂▂▂▂▂▂▂▂▂
Smoking cessation	45.33	1994	2006	▂▂▂▂ ▃▃▃▃▃▃▃▃▃▃▃▃▃ ▂▂▂▂▂▂▂▂▂▂▂▂▂▂▂
Stress	62.87	2019	2021	▂▂▂▂▂▂▂▂▂▂▂▂▂▂▂▂▂▂▂▂▂▂▂▂▂▂▂▂▂ ▃▃▃

The blue bar presents the time-period from the year 1990 to 2021, while the overlapping red color exhibits the period with citation burst activity.

**Table 4 ijerph-19-06128-t004:** Theories in patient education research yielding the highest centrality values (≥0.1).

Author, Year	Theory	Comment
Lazarus and Folkman (1984) [18]	Stress, Appraisal and Coping	A detailed theory of psychological stress, building on the concepts of cognitive appraisal and coping
Bandura A (1986) [19]	Social Foundations of Thought and Action: A Social Cognitive Theory	An influential piece of work in psychology that presented a comprehensive theory of human motivation and action. It explores the role of cognition, self-regulation, and self-reflective processes in improving psychosocial functioning
Ajzen (1991) [8]	Ajzen’s theory of planned behavior	Behavioral intentions and actions result due to interplay of three components: attitudes, subjective norms, and perceived behavioral control
Marlatt GA (1985) [20]	CB model of the relapse process	It challenged the then notions of viewing relapse as “treatment failure”
Woodcock (1990) [21] and McGrew (1991) [22]	Measures of Cognitive Ability	Laid down the theoretical foundations of -R measures of cognitive ability
David Baker (2006) [23]	Meaning and the measure of health literacy	Defining health literacy.

## Data Availability

All of the data supporting the findings of this review are included in this published article and Appendix A.

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
