# Peer review of "Evolution of Therapeutic Patient Education: A Systematic Scoping Review and Scientometric Analysis"

_ijerph, 2022, doi:10.3390/ijerph19106128_

Round 1
Reviewer 1 Report
- Therapeutic education is an important field of development of patients care. It is examined here through an original analytic technique that leads to interesting information. However, the analysis produced are more illustrative than demonstrative and should be accompanied by a more thorough discussion.
- The limitations of the work should be made clear. To my knowledge, the most important ones are those related to the definition of therapeutic patient education: the search used several keywords, which is quite understandable, but this results in the inclusion of works about patient information such as the domains of "polyvaccine information" and "mammography usage" that are clearly not TPE topics (or maybe it is the title of the paper that is not adequate?)
- Another built-in limitation is the restriction to English language which has more consequences in this particular field of medicine than in many others. For example, a lot of research and innovations about TPE were performed in Belgium, Canada, Switzerland and France, but even the main leaders recognized in those countries are scarcely cited. We feel this language restriction should be at least addressed in the discussion.
In addition to an expanded discussion, there is also a need for an improved explanation of the methods used which are not familiar to most readers (we are discussing about education, aren't we?).
Less importantly, there are some inadequations in the data presentation, e.g.
- Fig 1 : the figure suggests exponential growth of the literature rather than a "linear trend"
- Fig 2 : the addition of the four columns makes more than 100%, so is it really the proportion that is illustrated here?
Author Response
Please, see attached word document for detailed responses to comments.

Reviewer 2 Report
The document is well-presented and used Citespace to generate the graphs and tables. Citespace is well-known in scientometric analysis. The document has an important contribution to therapeutic patient education.
Author Response

(The authors gave the same response as above.)

Reviewer 3 Report
The article presents a timely and interesting review.
The state of the art is presented in detail with an adequate number of cited papers. The formatting of the article and English require only small modifications.
However, the methodology of the review process should be presented in more detail: the PRISMA guidelines should be used. The authors should incorporate them into the paper. http://www.prisma-statement.org
Also, the citations should be checked as it looks like that some of them are in APA style and not between brackets.
Author Response

(The authors gave the same response as above.)

Round 2
Reviewer 1 Report
I am satisfied with the answers and improvements made by the authors. This work is interesting indeed.